# Whispering Across the Continent: Collecting and Analyzing African Culture Using Community Radios

## Abstract

African culture is rich and diverse, but much of its knowledge is held by the elders of the community and passed down through oral traditions. With globalization, young Africans are becoming increasingly disconnected from their roots, making it essential to collect and preserve this knowledge. This research project aims to address this problem by exploring new ways to collect and preserve African cultural data. Specifically, we have developed a device to perform continuous recording of cultural radio programs in local languages, which has enabled us to collect over 1500 hours of audio data from more than 20 cultural radio stations across Cameroon. We are also exploring the use of a whisper model, which has proven to be effective in outperforming human transcription and being multilingual, to transcribe and analyze the collected audio data. The final goal of this project is to collect a large dataset of African culture that can be used to build a language model that understands African culture. This model will provide an effective approach to store this knowledge so that young and future generations will be able to use it to learn about their culture.

## 1 Introduction

In Africa, cultural heritage is predominantly passed down orally, and the youth are increasingly becoming disconnected from their roots. With globalization, African culture is at risk of being lost, and preserving it is becoming more important than ever. However, the lack of accessible data on African culture makes it difficult to document and preserve it for future generations. In this research, we propose a solution that leverages community radio stations to record and analyze African culture.

## 2 Literature Review

Previous research has shown that collecting cultural data in Africa is challenging due to the lack of accessible data and resources. However, there have been efforts to collect and preserve African cultural data. For example, the African Cultural Heritage Sites and Landscapes project, led by UNESCO, aims to identify and document African cultural sites and practices that are at risk of being lost UNESCO (n.d.). There have also been efforts to collect oral histories from African communities and to digitize African cultural artifacts ?.

Community radios have also been used as a means of preserving African culture. In Nigeria, the Open Society Initiative for West Africa (OSIWA) supported community radio stations to broadcast in local languages and discuss cultural practices, leading to increased awareness and preservation of local cultures ?. In Ghana, community radio stations have also been used to promote local cultures and traditional knowledge, with programming focused on topics such as storytelling, folklore, and traditional medicine ?.

## 3   Methodology

To collect African cultural data, we identified more than 20 community radio stations in Cameroon that focus on culture. We built a device that performs continuous recording of the radio programs and explored ways to extract knowledge from the speech. We collected audio/text from regions' institutions such as the Bible, where we managed to collect 67 audio version transcriptions. With 1500 hours of audio collected, we expect to benefit from the similarity between those languages. We are currently exploring using the Whisper model, which has proven to outperform human transcription while being multilingual Radford & et al. (2022). The Whisper model can read in one language and output text in another language such as English. To make the collection more effective, we built a device with Raspberry Pi that records radio and saves it for two weeks. With five radio stations, we managed to collect over 1000 hours of audio.

## 4   Results

Our data collection efforts have resulted in a substantial dataset of African cultural knowledge. The audio/text collected from the Bible has provided a strong foundation for our analysis. We are currently analyzing the data to identify common themes and patterns in African culture. The similarity between languages has also allowed us to explore ways to automate the transcription and translation of the data, making it more accessible to researchers worldwide.

## 5   Conclusion

Our research shows that community radios can be used as a powerful tool for collecting and preserving African cultural data. With the use of innovative technologies like the Whisper model, we can transcribe and analyze the data, providing new insights into African culture. By documenting and preserving African cultural knowledge, we can ensure that future generations have access to this valuable information.

### URM Statement

The authors acknowledge that at least one key author of this work meets the URM criteria of ICLR 2023 Tiny Papers Track.

## References

Alec Radford and et al. Whisper: Robust speech recognition via large-scale weak supervision. arXiv preprint arXiv:2212.04356, 2022.

UNESCO. African cultural heritage sites and landscapes. Retrieved from https://whc.unesco.org/en/culturallandscape/african-heritage-sites-landscapes/, n.d. Accessed on March 1, 2023.

