# OpenReview forum: "Whispering Across the Continent: Collecting and Analyzing African Culture Using Community Radios"
_ICLR.cc/2023/TinyPapers — Submitted to Tiny Papers @ ICLR 2023_

### Official Review · Reviewer_aPoH · 2023-03-18

**Confidence:** 5

**Summary Of Contributions:**

Paper lacks proof of work: IoT(Raspberry Pi) for audio collection and Whisper model for language translation.

**Rating:**

Needs Clarification (NC): a submission which does not meet the reviewing criteria and needs clarification for its described problem or solution

**Strengths And Weaknesses:**

Strengths:
1. Its admirable to build IoT device built with Raspberry Pi for audio collection
2. Utilize Whisper model for language translation
3. Written clearly for better understandability
4. Wanting to preserve African culture

Drawbacks:
1. No architecture/workflow of IoT device to Whisper provided
2. The paper seems to suggest whisper model did well in comparison, however no tabular information of the performance metrics for Whisper model is provided
3. I have worked with Whisper model(hugging face) from open AI python package, however there is no mention of the same apart from mention of the associated research paper
4. Currently paper seem to suggest its more of a speculation with very little research and very little Machine Learning implementation from their end.
5. On good faith, I can only assume little Machine learning was implemented with Whisper model. Paper doesn't currently provide any evidence on their work or research

**Suggested Changes:**

1. Do include the architecture of IoT device or diagram that provides the workflow from IoT to Whisper, this will enrich the paper
2. Do include a tabular information of the performance metrics used for Whisper model.
3. Do a performance comparison with different African languages
4. Authors should research more.
5. On good faith, I can only assume little Machine learning was implemented with Whisper model. Paper doesn't currently provide any evidence on their work or research

---

### Official Review · Reviewer_FPUo · 2023-04-02

**Confidence:** 4

**Summary Of Contributions:**

The paper shows that community radios can be used as a powerful tool for collecting and preserving African cultural data. Also, a device that performs continuous recording of the radio programs was built. The work is still in progress but the end goal is to curate a large dataset can be used to train a model on African culture.

**Rating:**

Needs Clarification (NC): a submission which does not meet the reviewing criteria and needs clarification for its described problem or solution

**Strengths And Weaknesses:**

Strengths
- Africa is known to be a low resource region in the NLP field as there very few resources available in languages from this region on the internet and the authors of this paper collected their datasets from radio programs in a very innovative way.
- Use of Raspberry pi to collect data
- The authors used whisper model to bridge the gap between the dataset they were able to get and the dataset they need.
Weaknesses
- Literature review didn't contain literatures where whisper model has been used to do something similar, even if it was used in a high resource setting.
-There are no results yet as the work is still in progress.
-The Bible is to be used alongside the dataset gotten from the radio stations to compare different African cultures. The bible dataset is a religious one and not a cultural one, so how it is going to be used to achieved this is not clear.
- Little details on how the data collection was done and also how the whisper model is being used to transcribe and analyse the data

**Suggested Changes:**

-I understand that the work is still in progress but the dataset that was collected was collected for only Cameroon, the authors didn't state if they have plans of collecting for other African  countries but the paper states that it's goal is 'to collect a large dataset of African culture that can be used to build a language model that understands African culture'. If the authors have no plans to explore outside Cameroon, would be Ideal for the work to be focused on Cameroon's culture instead of Africa's culture.
-The authors should clearly state what the purpose of collecting dataset from the Bible is, as the dataset is meant to be focused on culture and the Bible's domain is outside culture.
-The authors should include details on how the dataset was collected and also how the whisper model is being applied.

---

### Meta-Review · Area_Chair_t3hD · 2023-04-06

**Recommendation:** Invite to revise
**Confidence:** 5

**Metareview:**

1. Though the paper does provide information of other relevant literature, this paper fails to provide justification of their findings.
2. paper does not discuss about novel findings
3. Code & data aren't provided for reproducibility
4. Its a work in progress, with very little information provided
5. This paper does follow basic requirements however needs to revise to be CCR

**Summary:**

IoT device to record biblical audio/text from community radio with very little evidence

**Comments And Feedback To The Authors:**

The paper is a good start, you can improve on the paper.
Kindly revised the paper
1. provide more information on data collection techniques with architecture & workflow
2. Comparison with low and high resource setting with whisper model
3. metrics utilized to compare performance of human translation and whisper model in tabular format
4. Focus to collect data for more than one African languages, this can help to provide a performance comparison of different African language.

**Reason For Not Giving A Higher Recommendation:**

1. using whisper model as a ways to preserve African culture, IoT device built to record audio from radio
2. Text and Audio are collected from bible for their research
3. There is very little information how the bible is linked with the African culture
4. There is no information of how well it performs in comparison with human translation
5. No Information provided on whisper model transcribe and analysis done
6. Agree with reviewers point where bible is a religious book and not a cultural one


**Reason For Not Giving A Lower Recommendation:**

Agree with the rating and recommendations provided with both the reviewers.
This paper needs to be revised

---

### Decision · Program_Chairs · 2023-04-07

No revision received; not invited to archive